# Cross-Domain Indoor Visual Place Recognition for Mobile Robot via Generalization Using Style Augmentation

**DOI:** 10.3390/s23136134

**Published:** 2023-07-04

**Authors:** Piotr Wozniak, Dominik Ozog

**Affiliations:** Department of Computer and Control Engineering, Faculty of Electrical and Computer Engineering, Rzeszow University of Technology, Al. Powstańców Warszawy 12, 35-959 Rzeszow, Poland; d.ozog@prz.edu.pl

**Keywords:** visual place recognition, CNNs, multi-domain learning, domain generalization, transfer learning

## Abstract

The article presents an algorithm for the multi-domain visual recognition of an indoor place. It is based on a convolutional neural network and style randomization. The authors proposed a scene classification mechanism and improved the performance of the models based on synthetic and real data from various domains. In the proposed dataset, a domain change was defined as a camera model change. A dataset of images collected from several rooms was used to show different scenarios, human actions, equipment changes, and lighting conditions. The proposed method was tested in a scene classification problem where multi-domain data were used. The basis was a transfer learning approach with an extension style applied to various combinations of source and target data. The focus was on improving the unknown domain score and multi-domain support. The results of the experiments were analyzed in the context of data collected on a humanoid robot. The article shows that the average score was the highest for the use of multi-domain data and data style enhancement. The method of obtaining average results for the proposed method reached the level of 92.08%. The result obtained by another research team was corrected.

## 1. Introduction

Visual place recognition (VPR) is now a well-defined problem. In the last decade, the place recognition community has made many breakthroughs in loop closure detection [1] and simultaneous localization and mapping [2]. However, they constitute a problem that is extremely difficult to solve due to the wide range of implementations. In the context of visual recognition of a place, our source is an image. The basic question that needs to be answered is whether the place shown in the photo has already been seen. An additional challenge in VPR is that it can involve a human, a robot, or other machines. Depending on the environment and assumptions, conditions, such as the point of view, lighting conditions, or observed equipment, change, which requires additional handling mechanisms that are computationally expensive [3]. There are many already-proposed methods that perform well in static scenarios where we are aware of the changes taking place. However, they are not adapted to implement new solutions and test non-standard scenarios. An example of such an approach is the use of a humanoid robot. To perform more complex tasks, the robot must locate itself based on, for example, information from the image. An important limitation is the specificity of the robot and elements, such as the changing environment and cameras. Developing a working mechanism to recognize a place for a robot based on visual information alone is an extremely difficult and complex process. Such a device has computational and, especially, energy limitations. There are many risks associated with the use of a mobile robot in a real environment [4]. Apart from the standard problems that occur in the environment, the movement of a humanoid robot is burdened in a specific way, which causes a specific pose and motion blur from the camera. Gathering the data needed for site recognition by the robot is time consuming and expensive. Collecting data under other assumptions does not guarantee their positive application on a moving robot. The main effect of the changes is the deterioration of the work of the model developed for VPR in relation to the data from the test or implementation. In the literature, this issue has been mainly related to domain change [5]. The concept has been interpreted as changing the scenario or its assumptions. This is evident when the model does not generalize sufficiently to the new data. An example of such a situation is the use of a model to recognize a visual place in conditions that are considerably different from those of the training set. In the case of a robot, there may be different conditions at work. Taking into account multiple domains is important in the context of current heavily developed machine and deep learning algorithms [6]. Algorithms are available to deal with elements, such as a domain shift [7] or model overfitting. Additionally, there are practices that can be employed to mitigate the impact of a domain change. In the context of generalization, the following can be specified: data manipulation, representation learning, and learning strategy [8]. The natural procedure is to collect enough diverse data so that the developed model correctly handles unknown images. Additionally, analyses and preliminary compliance checks must be performed, for example, with t-SNE data visualization [9]. Unfortunately, this approach entails the need to collect and process more data. Data from other available collections, such as ImageNet [10], despite its complexity, do not allow for the direct implementation of new goals. Approaches, such as augmentation and data generation, have been used. An alternative to generalization is to try to adapt the domain [11,12]. Based on the new data, an attempt is made to adjust against the feature analysis. The attempt to solve the problem by generalizing and adapting has the support of the community. There are many classical or machine learning methods that are introduced. The use of data from multiple domains is a complex issue, and its solution depends on the specific situation. A common practice to better generalize the model is to train on multi-domain data. Based on the gap in the occurrence of the set in the context of a domain change, it was proposed to use data representing an attempt to use images in the VPR for a humanoid robot from independent sequences. Several models of cameras with different parameters were used in the research. The key information obtained in the work was the impact of the unknown domain model on its functioning. In addition, experiments were performed for the use of data from multiple domains. The question as to whether the VPR result could be improved with the help of data collection and manipulation was answered. The main focus of the work was the use of single or multiple sources as training data. The source was defined as a domain whose change is a problem for the work of the algorithms, especially when it is a change of data that are collected manually by a human to work on a mobile robot. Figure 1 shows a simplified diagram of the procedure. The scheme presents an attempt at visual recognition with the help of a single camera against the changing conditions, which are also unknown.

The lower, more elaborate variant was having multi-domain data collected. They represented the recording of separate sequences using different camera models. They came into contact with images from a wider spectrum of the domain. The focus was on observing the results for different scenarios: single domain and cross-domain [7,13,14]. The basic operation was the transfer learning mechanism, which was to change the weights of the convolutional neural network (CNN) [15] without much interference in the structure. The focus was on the input data and their impact on the precision of the solution. Based on the limitations of the moving robot, the operation of the algorithm was limited to the work of the CNN network. This article focuses on CNN validation in the context of VPR and several test situations. This document describes work that uses deep learning to recognize a room. The algorithm created focuses on the correct operation regardless of environmental factors, such as lighting, and regardless of the cameras used. In this way, the algorithm should work correctly, no matter what camera is used or when different cameras are used in different robots. This approach allows robot designers to use VPR on different hardware platforms and build swarms of different robots.

### Objective of the Work

Based on the literature review and publicly available datasets, issues related to the use of mobile robots were analyzed. The focus was on the visual recognition of the interiors of the rooms. Information on papers and datasets is included in the following chapters of the text. On this basis, the following shortcomings can be identified in the current research:a lack of a comprehensive multi-domain dataset for room recognition by the robot;a gap in knowing the results of the convolutional network for a wide spectrum of multi-domain data with environmental influence, e.g., human activity;a lack of analysis of the neural networks and methods for visual recognition scenarios for data from unknown domains and cameras;a lack of articles on preparing a swarm of robots for navigation based on significantly deviating data.

Based on the deficiencies described above, the main contributions of this paper can be identified as follows:the proposal of a domain extension approach using randomizing style for the problem of VPR;the testing of the approach for the proposed sequences from different camera models and on a real mobile robot;the improvement of the paper result of [4] with the proposed method.

The presented research leads to the development of an algorithm that can be used for a fleet of robots. The robots will differ in the ways they move, the cameras used, the height of assembly, and the angles relative to the ground. Despite the differences, the system will allow for room recognition for each robot independently. The assumption of the work is that one of the robots in the swarm will find itself in a new room, transmit information about it, and then each of the other robots will be able to recognize its presence in this space. In addition, the algorithm takes into account small changes that occur naturally in space, such as changing the position, appearance, or disappearance of small objects, moving used objects like chairs, and changing lighting conditions or image fragments, such as the view outside the window. The challenge of the work is that the room should be recognized after it is photographed once by the first robot, which forces the authors to minimize the amount of data in the training sets. Extending the set with a very large number of photos would be contrary to the assumptions of the target system in which the algorithm would be running. The assumptions mean an increase in the difficulty of the problem, but also entail an attempt to reduce the future expenditure of future work.

## 2. Relevant Work

There are many challenges and methods for visually recognizing places. Initially, manual descriptors, such as GIST [16] and SURF [17], were used to solve this. With the development of deep learning, many neural networks were introduced that could be adapted to a particular problem, such as outdoor/indoor VPR [18]. The main role mechanism that obtained the description of the scene in the image was the deep convolutional network. Based on a trained structure, such as AlexNet [19], a feature vector describing the image was obtained. The community focused on proposing its own solutions or modifying existing ones [20,21]. Most of them required additional processes and increased efficiency to improve the result. The cost was disproportionate to the additional task of locating the robot. CNNs, such as GoogLeNet [22], ResNet-50 [23], and VGG-16 [24] in an unmodified form, became a standard and allowed for image analyses of a higher quality than classic methods. They also allowed for a more precise analysis of image fragments. The detection and recognition of objects [25,26] or segments [27,28] could be distinguished. However, the methods remained vulnerable to many difficulties. These difficulties also occurred in terms of the visual recognition of places. Their disadvantages include computational complexity and a greater need to refer to GPU resources. It has become popular to use a pre-trained network for feature extraction or transfer learning [29]. Extracting features from higher-level network layers has become a popular mechanism, for example, in image comparison. Descriptions are used to find the closest matching scene on the image. They become useful for working with lighter models, such as support vector machines (SVMs) [30], naive Bayes classifiers, and the K-nearest neighbors algorithm. Approach scores tend to drop when the data is not adjusted to supervised or especially unsupervised training [31]. The community focused on augmentation by generating data to better prepare the already-overtrained deep nets. It was important to confirm that the given manipulation would not worsen the results globally. In this decade, many modifiers have become available to enrich the training data, from basic rotations and perspective modifications, to adding noise or through effects based on deep methods. This is a way to improve results in simpler scenarios with a higher percentage effectiveness [32]. It is an encouraging prospect for further work in this direction. These types of approaches have focused on working with robots or mobile vehicles that are trying to recognize a scene.

An important assessment of the proposed parameters is the collection of test data. In the literature, many papers and datasets on visual navigation [33] and robot navigation in internal and external spaces can be found. Images that show scenes in different conditions and seasons over long periods of time affect the data domain and make it difficult to obtain a satisfactory result using the methods. An example is the collection of Nordland [34]. As with TokyoTM [35] and Pitts30k [35], it presents complex scenes in various variants, e.g., day and night, on a large scale. An extensive collection that includes classifications of various rooms in various conditions is Places365 [36]. The presence of over 10 million different images from 434 types of rooms considerably enriches the initial preparation of models and their testing. However, the mentioned data do not represent the scenario of the robot working in an inside space. In this paper, the authors present two variants that inspired the research. In the first variant, the camera was mounted on the robot. In the second, the camera was moved between rooms by the researchers. Then, the algorithm determined the location of the device based on visual information. In the literature, there are works in which the robot moves around different rooms [37,38,39,40]. The works analyzed the change of the robot, its pose and, to a small extent, the parameters of its movement. When the recording camera is changed, the difficulty lies in introducing further environmental conditions and the robot itself. The basic limitation of the mentioned sets is the lack of reference to changes in the recording device. A modification of the ambient conditions alone is only a partial change within the domain. The biggest gap of the presented work is the failure to take into account the situation of changing the robot and the camera. In addition, a situation in which a model is prepared from a completely different domain was not considered. It is natural for a human to capture images and transmit them to a robot, drone, or other hardware platform.

## 3. Dataset and Method

The algorithm is based on accepting the limitations of the mobile robot in the processing of advanced images. The focus was on the initial preparation of the data and the model working on the robot. The most important aspect was data manipulation, which would have improved the generalization of the model. No additional rules or mechanisms supporting the classifier were defined during training. Therefore, it can be concluded that the input data and the training procedure were the most important elements of the paper. Figure 2 shows the general scheme of the algorithm. Its elements are discussed in the following sub-chapters.

### 3.1. Multi-Domain Dataset

The main task was to propose a dataset with a variety of sequences. Changes in lighting, equipment, and human activity coincided with problems that can occur in reality. The expanded dataset included images from the article [41] and new proposed subsets. The subsets focused on the issue of room categorization in sequences and various conditions, e.g., blur and deblur. Image sequences were recorded for nine different locations by a humanoid robot. The Nao robot V4 manufactured by Aldebaran Robotics with OpenNAO software had an Intel Atom Z530 @ 1.6 GHz processor, 1 GB RAM, and 8 GB Flash memory. The Xtion PRO LIVE camera was placed on it to capture images of the environment. The camera worked in VGA standard, with a resolution of 640 × 480 and shooting at 30 fps. Information from the robot’s sensors was limited to visual information, i.e., a color camera. The subsets added were based on different camera models. In Figure 3 images from different cameras and under the influence of changing conditions are presented.

Despite the visual similarity from the perspective of human interpretation, current algorithms are susceptible to visual differences. The change may seem difficult to detect without prior analysis of the data. It is worth noting that, depending on the test, the focus was on introducing various difficulties. The Test 2 sequence is particularly noteworthy in that the activity of people on the stage was also introduced. The division of each of the subsets was made up of a training part and three test parts. These were separate passages through the rooms. This facilitated a more accurate assessment of the impact of individual disturbances. The exception was the first set of the robot (Xtion Camera), which did not contain the Test 2 sequence. There were nine of the same rooms in each sequence: three corridors and six different university laboratories. The number of images assigned to each of the subgroups varied due to the characteristics of moving around the rooms. The sequences represented different domains depending on the cameras used and the conditions. An example of the same scene in separate domains is shown in Figure 3. In some cases, interpreting a source from just a single image was difficult. Within a subgroup, each of the sequences was balanced. Each room was represented by the same number of photos. The general breakdown is presented in Table 1.

Differences in the resolution and aspect ratio of the saved images were also presented, which also affected the final results. The key information to recognize the problem of moving inside rooms was what elements may have been present in the environment. The proposed collection included various types of rooms: corridors, research laboratories, and larger laboratory rooms. The camera, on a robot or held by a hand, could be burdened with motion blur. This was especially visible for a humanoid robot regarding changes in indoor and outdoor lighting conditions (time of day). The collection also focused on human activity. Depending on the room, single people or groups of people could perform on stage. Changes in time also had an impact on the interpretation of the scene. An important difference in the context of the robot application was the difference between sequences without the use of a robot and with a robot. The difference was in the pose of the camera and the way it moved. Both approaches often made the scene visible from a different perspective, which made VPR much more difficult. Examples of disturbances in the stage are presented in Figure 4. In the initial analysis, differences in the images from different cameras could also be observed. In RGB images, the difference can be difficult to interpret and is more visible in the HSV format, as shown in Figure 5.

The data were not subjected to additional modifications in terms of contrast, color saturation, blurring, and noise reduction methods. This was due to the need to test the method on raw data. Adding further mechanisms would also have made it difficult to run the method directly on the robot. The initial selection of cameras and their configuration would have reduced the difficulty of the task at work.

### 3.2. Style Randomization

The authors in [32] multiplied the number of domains in a synthetic way based on original images. There are many classical data augmentation solutions and newer deep methods available in the literature [42,43]. The present process uses CNNs to freestyle the image and combine it with others. Such a process is called neural style transfer (NST) [15]. This article uses the augmentation code [32]. Figure 6 shows an exemplary operation of the algorithm. This modification is referred to in the work as style augmentation, and it is the basis for a synthetic change of the image domain from the dataset. Unlike a simple augmentation method such as noise, rotation, or translation, style changes more strongly affect textures without changing the geometric information of the image. This is often expected when interpreting scenes where the main feature is the arrangement of walls and equipment.

In our experiment, the default algorithm for the style randomization method was used. Configurations were used according to the work of [32]. The disadvantage of the approach in relation to the classical noise methods is the computational complexity. Augmentation on the robot during its work was too demanding. This process was carried out as a stage of data preparation before deep network training.

### 3.3. Transfer Learning

The standard approach in machine learning methodology is that the training data and the test data come from the same domain. However, some situations require a different approach, and it is very difficult to maintain this rule. In the case of VPR and the use of a robot, collecting new data is time consuming and often technically problematic. This is especially difficult when large amounts of data are needed. In the case of deep learning networks, this is often the basis. The approach referred to as transfer learning [29] is helpful. In this method, a learned model is applied to another related problem. The goal is to obtain a result for the data from a new domain. In this approach, the size of the new set is less important. The method works well for CNNs and is further explored in the article. The work used MATLAB to train deep networks with the Deep Learning Toolbox [44]. The tool allowed access to trained networks and scripts to perform transfer learning. During training, the last layers of the ResNet-50 network were modified to adapt them to the problem of room classification. Classification and fully connected layers were added, while the weight learn rate factor and bias learn rate factor were set to 2. Training the data depended on the test variant. Apart from the unmodified training sequences, standard augmentation (std. aug.) was used. The image was changed randomly by modifying rotations from −10 to 10 degrees and translations in the X,Y axis from −10 to 10 px. The proposed algorithm from one image was extended to four new images with separate random domains. An example image subjected to style randomization is shown in Figure 6. The operation was part of the preparation of the dataset. Two training epochs were used for a single training sequence, and three max epochs were used for a group of sequences. The stochastic gradient descent with momentum optimizer was used for all CNN training in the work.

The limitation of the transfer learning process [45] lies in the fact that the process does not meaningfully affect the structure of the network. Some of the last layers may be modified, but, in the standard approach, it does not considerably affect the result. When the initial result for weights, e.g., from the ImagNet dataset, is not high, there is little chance of a major improvement in the relative approach. It is necessary to completely rebuild the approach or change the network. In the event that the primary result is accepted, an improvement is possible on the new data. The current problem, however, is the collection of new data or their extension. The incorrect selection of the augmentation method may cause a deterioration in the result. The learning process is also computationally demanding for more complex CNN structures.

### 3.4. Features Extraction

Mobile robots are burdened with computational limitations. In addition, in the assumptions of this paper, they do not use external calculated units, e.g., cloud resources. Therefore, the focus was on preparing and optimizing the model for a specific implementation. The dataset processing operations, such as archiving, augmentation, CNN training, and SVM modeling, were not performed on the robot. It was too computationally expensive in conjunction with the work of the robot. An external computer adapted to these tasks was used. The images were processed, the data were normalized, and the networks were trained on the computer. Five structures of the CNN network were used in the work, one of which was modified for the process of transfer learning. We distinguish the following structures:ResNet-50 [23] consists of 177 layers and was trained on the ImageNet dataset [46]. In the experiments, it was modified and trained on multi-domain data. The fully connected layer was modified to the number of rooms. Information about the changes can be found in the transfer learning chapter. The diagram of the structure is shown in Figure 7;VGG-M [47] consists of 21 layers, was trained on the ImageNet dataset [46], and was used for the ImageNet Large Scale Visual Recognition Challenge of 2012 [48];GoogLeNet [22] consists of 144 layers and was trained on the Places365 dataset [49]. These CNNs learned different feature representations for a wide range of images. The focus was on the visual recognition of the place in the interiors of the rooms;VGG-16 + NetVLAD consists of 35 layers, including its NetVLAD Layers. It was trained on the TokyoTM dataset [35].

The prepared models ware implemented on the robot. A useful solution in the context of collecting new data was the extraction of features [50] from the deep network. The newly registered images were classified, and their representation could also be saved. When choosing the network, not only was its size important, but also its output vector. Table 2 shows the network layers from which the image feature vector information was obtained.

Information about network weights and layer name was included in this work. When storing a value, a feature vector that is twice as long means that the available space is halved. This had to be taken into account in the context of the difference in results. Unlike the processing speed of a single image, this was critical. VGG-M [47], as well as VGG-16 [24] + NetVLAD + whitening [35] (this method is called VGG-16 + NetVLAD later in the article) required the most memory for one image. The smallest feature vector was used with GoogLeNet. In this work, the processing time of a single image for each network structure was checked. The batch size was set to a single image that was similar to the real work in the environment. In the test, 1000 images were processed according to the following chapters. The image processing time was averaged, and the standard deviation was determined. The results for the use of CPU and GPU are presented in Table 3. The network processing time was the most time-consuming element handled by the network, especially for use on a robot without GPU support. The image pre-processing time and SVM model operation time were not analyzed. This was due to the low computational requirements of these operations. The times obtained were for the given specification. The detail device specifications include the following:Processor (CPU): AMD Ryzen 7 4800H;Graphic card (GPU): NVIDIA GeForce GTX 1660 Ti;RAM memory: 16.0 GB DDR4.

## 4. Experimental Results

The experiment was divided into two main parts. The first assumed that the changes would be made only within one domain of the recording device. The chapter covers two sets and the most popular deep learning methods available. The focus was on fundamental issues, such as changes in lighting, equipment, and human activities. The aforementioned problem was considered as a basis for testing whether training with additional images with a synthetically modified domain would improve the result. The conditions of each test were very similar and focused on training the CNN to extract a feature from it. Then, a second SVM model was trained from the received descriptions. The CNN training data and the support vector machine model did not include images of the test sequences. The measure of obtaining the method was the accuracy in determining what parts of the images were correctly classified into the room. The SVM parameters in the “One vs. All” mode were not changed within the individual tables. The properties were fixed at the beginning and were mainly based on default values from the MATLAB environment. Using image descriptions on unchanged SVM properties allowed for the evaluation of the quality of the CNN work. The hyperparameters of the second model in the article remained unchanged.

In this work, standard assessment indicators were used, with the basic ones being accuracy, precision, recall, and F1-score [51], which represent information about misclassified rooms from the sequence. When tested on a group of test sequences, the result was averaged, and the standard deviation was reported. For the tables, an additional boxplot [52] presenting the distribution of statistical features was added to visualize the median, maximum, and minimum values, as well as the outliers for the methods.

### 4.1. Single-Domain Recognition

In the first stage, the focus was on the basic room classification scenarios. Camera compatibility was maintained in the context of training and test sets. The only changes were environmental problems introduced in the test sequences, e.g., a change in lighting conditions.

The experiment aimed to compare the selected deep networks in terms of their application to VPR. An important element was to train one structure, i.e., ResNet-50, and check whether the proposed approach would improve the result. The basic network weights obtained after training for simple augmentation and for the proposed method were included. The results looked to answer the question of how existing deep networks deal with the data collected. In addition, the influence of augmentation on the final results was checked. After preparing the models separately, using training sequences, their performance was tested in related test sequences. Table 4 and Figure 8 present the experiment’s result. The research used a wider range of different deep networks. This allowed for the initial selection of structures for further research. VGG-M (ImageNet) [46] and GoogLeNet trained on the Places365 [36] dataset were tested. The focus was mainly on ResNet-50, and columns with results for three variants of its weights have been introduced in the results. The first is the weights obtained on the ImageNet set, and the second is the results after training based on documented data. The third variant was the proposed method, which used synthetic data on the changed domain applied by the method. The ResNet-50 network was selected due to the high initial network score. For the baseline weight obtained from ImageNet training, the method had an average score of 88.95%. For training on training data, it reached 90.71%. The VGG-16 network with the NetVLAD modification obtained a score of 91.07%. The highest average score was obtained after adding and training the ResNet-50 model with additional ones added from new domains. The result was equal to 91.60%.

The second stage of the basic classification was to check the solution in other data. Reference was made to the results obtained in the paper [4]. The study again referred to and overtrained the ResNet-50. All network training sequences from the proposed set were used—this included four sequences (Nao extended). After retraining the network weights with images from nine rooms, the SVM network was reused. The test combination was similar to the works of [4,53]. The results of the model training with the use of multi-domain images are included in Table 5. The reason for such an operation was the similar type of data, and the network weights were in no way dependent on the test data. An improvement in the score to 98.74% indicated not having to use data from the deployment environment. The ResNet-50 model used was subjected to transfer learning based on images from the extended set proposed in the work. There was no connection between the rooms registered in the training and test sets [53]. The augmentation and training procedure were identical to those of the other configurations. In the case of using multi-domain data from similar data, it was possible to obtain a representation of features, thereby offering a better result. The test confirmed the stability of the results. The CNN ResNet-50, after proper training on multi-domain data, improved the representation of the image and the average final result. It did not require additional image blur detection mechanisms or additional augmentation at the level of the second SVM model. Most importantly, the training data for the CNN did not necessarily have to come from a dataset from the paper [53]. The network was trained using the training sequences in this article.

### 4.2. Multi-Domain Recognition

The second stage was checking when the consistency of the source and target domains was not maintained. This type of change means a deterioration of the result. This is presented in Table 6 and Figure 9. The average results with standard deviation from the ResNet-50 and VGG-16 + NetVLAD networks are shown. The proposed method improved the result for the classification, although a deterioration in the standard deviation could be observed. This was due to difficulties in test sequences 2 (Test 2) with human activities. The table shows the differences for individual cameras. The hardest combination was Xtion vs. the others.

The final step was to see if using cross-domain data from real cameras would improve the results. Experiments were limited to four domains from separate cameras. Different combinations of training sequences were used in the CNN and SVM training, and markings for connecting training subsets were introduced, as detailed in the following: #Set1—all excluding Xtion, #Set2—all excluding Iphone, #Set3—all excluding GoPro, and #Set4—all excluding P40Pro. With the exception of the #Set5 test, in which all training data were used, one domain was excluded in each study.

Based on the results from Table 7 and Figure 10, it is possible to assess the dependencies with respect to the combination of various data.

The lowest results were obtained when testing sequences from the robot, and such images were not used in training. The highest score in this group for the three methods with synthetic domains was 74.86%. The three subsets collected without robot activity were the closest to the results. This was due to the closer pose during the recording, but also to a smaller difference in the camera model.

The highest average result was obtained using the method based on ResNet-50 and a combination of all augmentations. Using an additional one to train on synthetic data improved the average score by two percentage points. The best results from the VPR perspective were achieved by using all available data and modifiers. The work attempted to visualize the features obtained from the CNN and link them with the results obtained. The t-SNE method was intended to visualize the quality of the extracted features and the impact of changes in network weights. The diagram presents a layout with different color-coded rooms. The subsection contains the most interesting case, and the relationship between the data from the robot and other sequences is shown in Figure 11 and Figure 12.

Points represent training data from three recorded sequences and crosses from an unknown domain, i.e., a robot (Xtion). Visualization refers to the result obtained by ResNet-50 before transfer learning (64.14%) and after (74.86%). Improvement of the result was based on the use of synthetic and real images. A particular increase can be seen in Table 6. Separately applied sequences of different transitions did not provide a better result.

## 5. Discussion

Although VPR is a well-defined problem, the current methods of solving it are limited in terms of assumptions. Each extension of the range of location recognition entails a large amount of work, especially in relation to robotics applications. Collecting data, processing them, and creating a model is a time-consuming process that poses a major challenge when preparing a fleet of robots that may differ in specifications. The only constant in the task is that the robots move in the same rooms. As part of the proposed mechanism, the focus was on checking how the current deep networks would behave under different conditions. Particular attention was paid to changing the camera model. In addition to additional problems, such as changes in lighting conditions, changing the recording device was the biggest one. The result of changing the data domain was poor performance for the initial approaches. The proposed approach allows for the improvement of recognition results for various modifiers, starting from camera change, through conditions, and human activity. The applications of domain extension data augmentation demonstrated positive results in the case of real and synthetic images. This impact was assessed using data collected from various cameras, including a robot. The advantage of the approach based on adding data from unknown domains for VPR is interference in network weights. The approach outside of structure training does not require much interference in the classification mechanism. It allows for the subsequent introduction of additional mechanisms, regardless of the improvement of the result from the proposed approach. Depending on the constraints on the robot used, it is possible to use a common model trained on an external device. The high-precision method, despite the computational complexity, will reduce unnecessary data processing and free up resources for the robot’s work.

### Future Work

The proposed set and approach allowed for verification of the difficulty of creating a multi-menu mechanism for VPR. It was possible to pre-estimate the results for the application of deep learning. The results showed the need for further work on the topic. This is a prerequisite for the efficient creation and use of a fleet of robots in indoor conditions. The recognition of the place by the robot, regardless of obstacles and changing the camera, is a milestone for further breakthroughs. The presented results provide an impulse for further experiments in terms of the use of dedicated VPR augmentation. Improving the network score by applying multi-domain data and changing the weights means that the problem can be solved by optimizing the network parameters. This direction allows for the determination of how much of the problem can be solved by interfering with the network, and it allows for later confrontation with completely different methods from CNNs. Currently, we can observe great progress in the field of robots, which are designed to support the daily work and tasks of people. To perform their tasks, these robots not only have to be designed appropriately, but also have to function in the working environment of humans. This coexistence in one environment raises several problems related to the anthropomorphization of robots. One of them is the ability to recognize the room where the robot is based on the image from the camera. The authors in this document included a description of the work using deep learning to recognize a room. The created algorithm focuses on correct operation regardless of environmental factors, such as lighting, and is intended to function regardless of the cameras used. In this way, the algorithm should work correctly no matter what camera is used or when different cameras are used on different robots. This approach allows robot designers to use it on various hardware platforms and to build swarms of different robots that exchange information to recognize their location in those places of the building where only one of the swarms of robots was previously present. For this reason, the algorithm must learn to recognize rooms without having a very large number of photos of each of them (a thousand or more), and it should achieve the best results based on the number of photos that can be taken during a short robot ride around a given room (10–20 photos). In addition, the algorithm takes into account minor changes in rooms that have a negative impact on the operation of classic classifiers. These are changes, such as lighting depending on the time of day, the view outside the window, and the displacement or appearance of new everyday objects, such as books, notes or the position of chairs that have been changed. The work planned for the future is aimed at further developing the algorithm for cooperation in the human environment by taking into account the presence of moving people in a room and covering key areas that are characteristic of this room and the process of its recognition. However, a different data set will be created for this purpose, which takes into account the presence of people and their natural activities.

## 6. Conclusions

In this paper, we presented a complex dataset for multi-domain VPR. The collected images came from different sources, including the camera of a humanoid robot. We proposed a VPR algorithm for multi-domain and cross-domain data. We have shown that extending the data with new synthetic and real domains improves performance compared to available algorithms. A relatively simple, pre-trained CNN was used to test the domain adaptation approach. An important goal was to try to improve the classification score for unknown data by using better generalization through the network. This will allow the use of redundant domains, i.e., extended and separately collected images. It was assessed that the use of multi-domain data improves model performance in the context of completely new and distinct data. It had a positive effect on the performance of more standard environmental problems, e.g., changing lighting. The proposed domain extension approach improves accuracy even for new and different source data. In the case of a large change, the approach produced acceptable results.

## Figures and Tables

**Figure 1 sensors-23-06134-f001:**
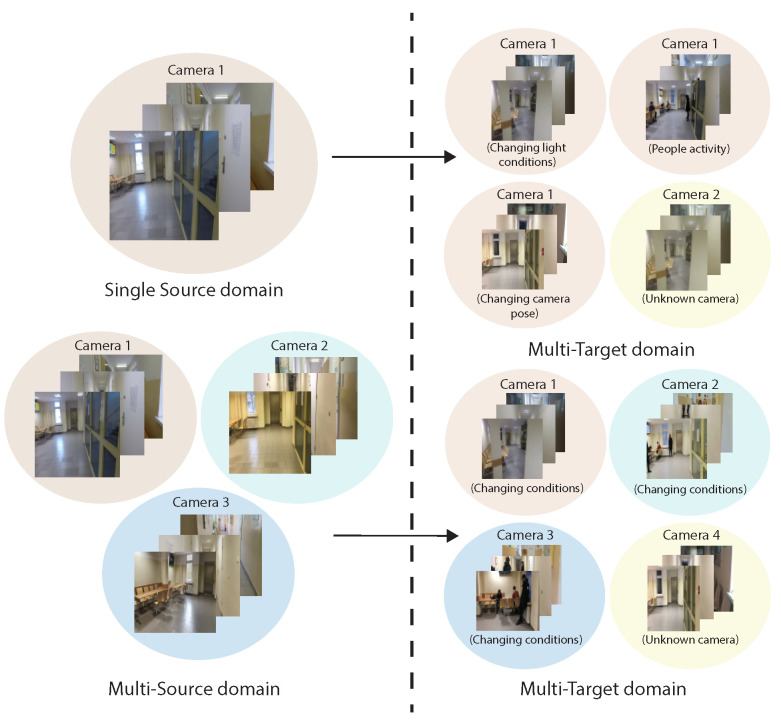
Scheme of a single domain or multiple domain usage scenarios.

**Figure 2 sensors-23-06134-f002:**
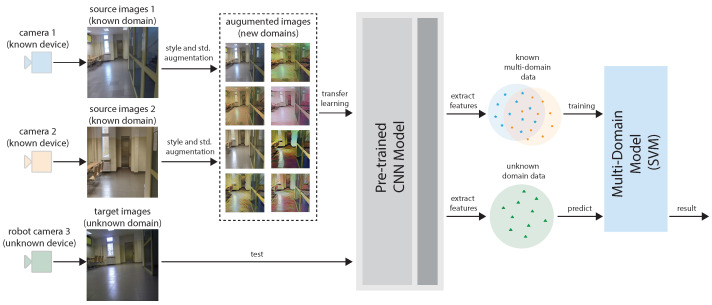
Algorithm diagram for cross-domain visual room recognition.

**Figure 3 sensors-23-06134-f003:**
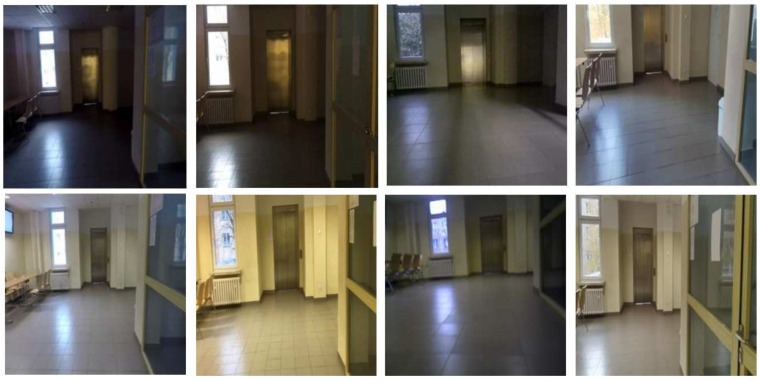
Example of images coming from different domains—up and down pairs of the same model (camera shots).

**Figure 4 sensors-23-06134-f004:**
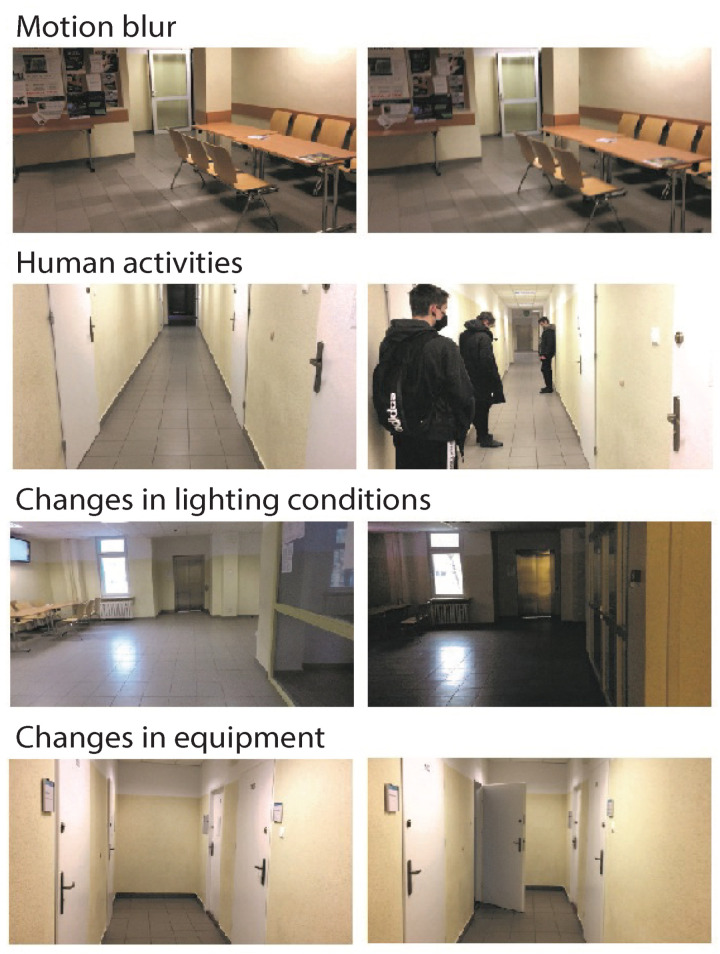
Example of problems in the extended dataset.

**Figure 5 sensors-23-06134-f005:**
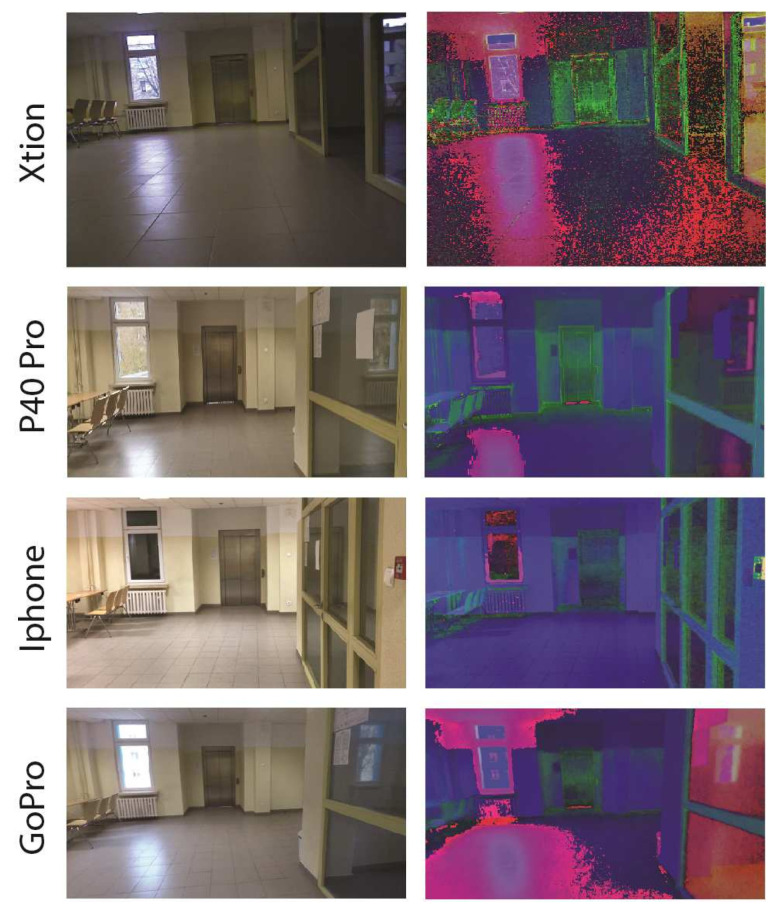
Visualization of the difference between the images—RGB (**left**) and HSV (**right**).

**Figure 6 sensors-23-06134-f006:**
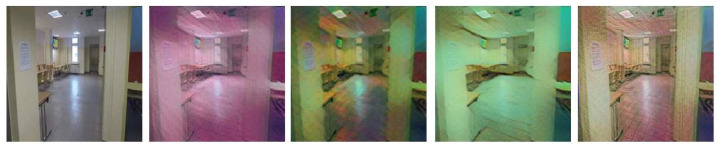
An example of a random image style from the data set.

**Figure 7 sensors-23-06134-f007:**
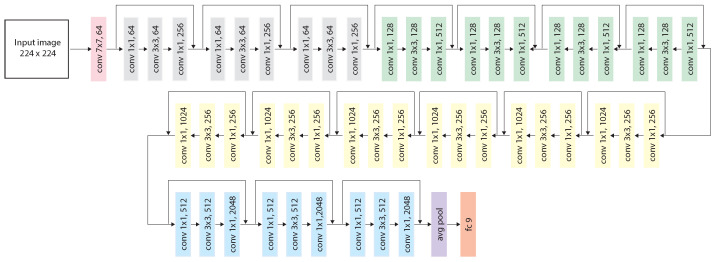
ResNet-50 architecture after transfer learning.

**Figure 8 sensors-23-06134-f008:**
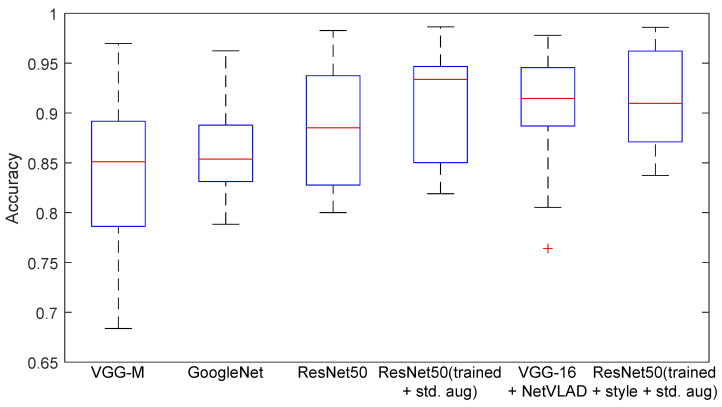
Diagram of method classification results for a dataset sequence.

**Figure 9 sensors-23-06134-f009:**
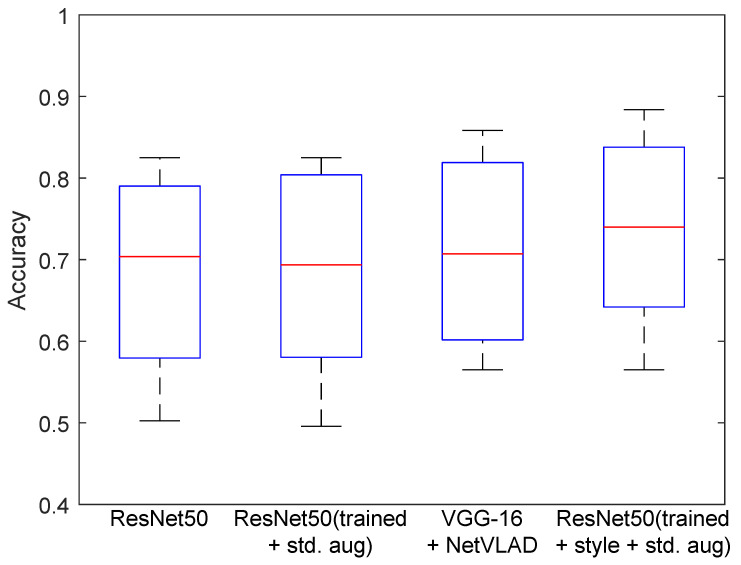
Diagram of method classification results for the different domain sequences of the datasets.

**Figure 10 sensors-23-06134-f010:**
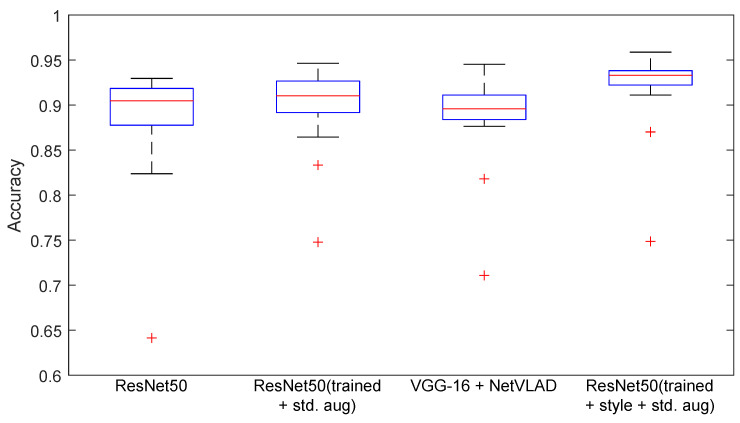
Diagram of sequence classification results using cross-domain data.

**Figure 11 sensors-23-06134-f011:**
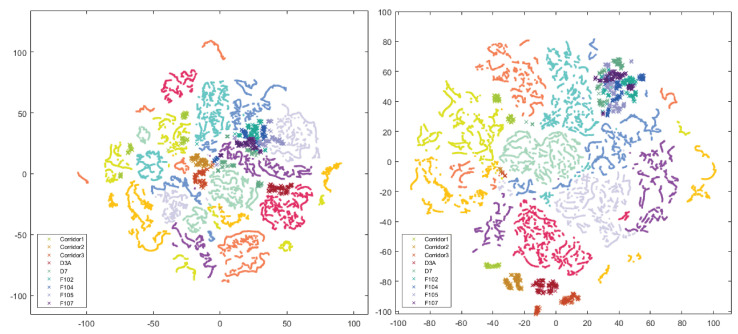
Visualization of features from all sets without Xtion vs. Xtion, VGG-16 + NetVLAD (**left**), and ResNet-50 ImageNet (**right**).

**Figure 12 sensors-23-06134-f012:**
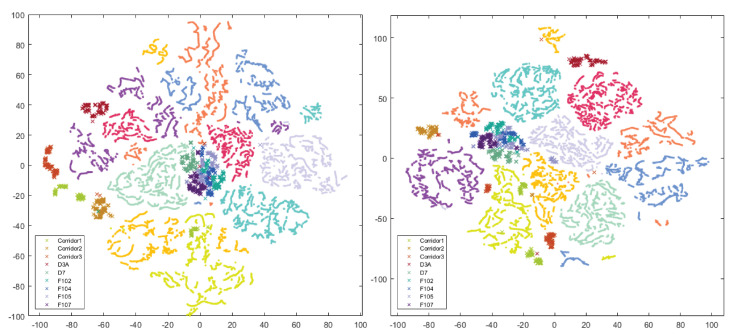
Visualization of features from all sets without Xtion vs. Xtion, ResNet-50 trained std. aug. (**left**), and ResNet-50 trained std. + style aug. (**right**).

**Table 1 sensors-23-06134-t001:** Properties of image and number for individual sets, subsets, and sequences.

Subset	Images No (Seq.)				Image Properties		
	Training	Test 1	Test 2	Test 3	Width (px)	Height (px)	Proportion
Xtion [41]	800	200	-	200	640	480	1.3
GoPro	600	500	500	500	960	540	1.7
Iphone	600	500	500	500	960	540	1.7
P40Pro	600	450	350	350	640	360	1.7

**Table 2 sensors-23-06134-t002:** Properties of the feature vector extraction of the deep networks used.

Model	Weight	Net Layout	Feature Size
ResNet-50	ImageNet	avg_pool	2048
VGG-M	ImageNet	relu6	4096
GoogLeNet	Places365	pool5-7x7_s1	1024
ResNet-50	ImageNet -> Trained	avg_pool	2048
VGG-16 + NetVLAD + whitening [35]	TokyoTM	VLAD vector	4096

**Table 3 sensors-23-06134-t003:** Time durations of the feature vector extraction of the deep networks used.

Model	Time (s)	
	CPU	GPU
Resnet-50	0.1461 ± 0.0072	0.1373 ± 0.0098
VGG-M	0.0561 ± 0.0055	0.0536 ± 0.0055
GoogLeNet	0.0915 ± 0.0118	0.0695 ± 0.0113
VGG-16 + NetVLAD + whitening [35]	0.2552 ± 0.0213	0.1607 ± 0.0068

**Table 4 sensors-23-06134-t004:** Classification results for the sequences of the dataset. The best score for the methods is marked in bold.

		Accuracy (%)					
**Source**	**Target**	**VGG-M**	**GoogLeNet**	**ResNet-50**	**ResNet-50**	**VGG-16 +**	**ResNet-50**
**(Training Seq.)**	**(Test Seq.)**				**(Trained + std. aug.)**	**NetVLAD**	**(Trained + Style + std. aug.)**
Iphone	Test 1	84.15	85.33	92.53	93.46	90.28	**94.46**
	Test 2	80.68	82.64	86.26	87.46	**91.46**	84.93
	Test 3	77.08	78.84	81.08	84.20	80.53	**90.35**
GoPro	Test 1	96.97	96.24	98.28	**98.64**	97.80	98.60
	Test 2	85.11	86.73	88.51	90.64	**91.31**	86.86
	Test 3	86.20	85.37	86.28	**93.37**	92.40	90.97
Xtion	Test 1	90.61	89.38	93.66	94.83	95.27	**96.11**
	Test 3	77.94	80.05	80.00	82.33	**88.16**	87.83
P40Pro	Test 1	89.33	89.65	96.49	96.79	95.40	**97.48**
	Test 2	68.38	**84.60**	81.61	81.90	76.41	83.74
	Test 3	88.69	86.98	93.77	94.19	91.77	**96.25**
Mean		85.40	85.98	88.95	90.71	91.07	**91.60**

**Table 5 sensors-23-06134-t005:** Classification results for the dataset in [53]. The best score for the methods is marked in bold.

Method	Accuracy	Precision	Recall	F1-Score
VGG-F fine-tuned [53]	97.19	97.12	97.20	97.16
Baumgartl et al. [4]	97.95	97.97	97.95	97.95
VGG-16 + NetVLAD	98.14	98.11	98.06	98.00
ResNet-50	98.60	98.50	98.55	98.50
ResNet-50 (trained + style + std. aug)	**98.74**	**98.73**	**98.66**	**98.68**

**Table 6 sensors-23-06134-t006:** Classification results for the different domain sequences of the dataset. The best score for the methods is marked in bold.

		Avg. Accuracy (%)			
**Source**	**Target**	**ResNet-50**	**ResNet-50**	**VGG-16**	**ResNet-50**
**(Training Seq.)**	**(Test Seq.)**		**(Trained std. aug.)**	**+ NetVLAD**	**(Trained std. + Style aug.)**
GoPro	Xtion	53.08 ± 1.75	58.86 ± 6.31	61.67 ± 4.67	**63.69 ± 4.81**
	Iphone	79.97 ± 2.65	82.00 ± 2.27	**85.84 ± 3.37**	82.72 ± 5.37
	P40Pro	78.62 ± 6.00	**80.19 ± 4.01**	77.03 ± 3.25	79.37 ± 7.60
P40Pro	GoPro	79.42 ± 2.63	80.59 ± 4.16	84.47 ± 4.10	**88.38 ± 4.90**
	Xtion	50.25 ± 0.08	49.58 ± 5.03	60.30 ± 3.97	**63.75 ± 2.64**
	Iphone	82.50 ± 1.63	82.50 ± 2.68	84.70 ± 1.43	**86.36 ± 3.10**
Iphone	GoPro	75.07 ± 9.21	74.26 ± 8.04	79.33 ± 8.76	**84.84 ± 7.51**
	Xtion	52.53 ± 9.47	52.02 ± 13.64	**56.50 ± 8.50**	56.50 ± 10.34
	P40Pro	75.11 ± 5.10	76.62 ± 6.44	79.11 ± 4.62	**80.29 ± 5.79**
Xtion	GoPro	64.96 ± 5.29	64.18 ± 6.98	64.39 ± 7.22	**68.62 ± 7.19**
	Iphone	65.67 ± 2.15	64.47 ± 3.40	60.02 ± 2.91	**68.42 ± 5.06**
	P40Pro	62.79 ± 2.15	57.23 ± 7.57	59.88 ± 8.79	**64.64 ± 12.18**
Mean		68.33 ± 4.01	68.54 ± 5.13	71.10 ± 5.13	**73.96 ± 6.37**

**Table 7 sensors-23-06134-t007:** Sequence classification results using cross-domain data. The best score for the methods is marked in bold.

		Avg. Accuracy (%)			
**Source**	**Target**	**ResNet-50**	**ResNet-50**	**VGG-16**	**ResNet-50**
**(Training Seq.)**	**(Test Seq.)**		**(Trained std. aug.)**	**+ NetVLAD**	**(Trained std. + Style aug.)**
#Set1	Xtion	64.14 ± 0.364	74.78 ± 3.23	71.08 ± 6.03	**74.86 ± 3.36**
	Iphone	91.86 ± 2.61	92.44 ± 3.83	88.26 ± 2.22	**93.40 ± 3.85**
	P40Pro	92.29 ± 5.31	**94.01 ± 3.48**	89.39 ± 6.78	93.88 ± 5.75
	GoPro	91.83 ± 4.76	93.54 ± 4.28	93.22 ± 3.64	**94.70 ± 4.20**
#Set2	Xtion	89.17 ± 4.67	86.44 ± 7.83	91.00 ± 3.45	**92.19 ± 3.81**
	Iphone	89.93 ± 0.85	89.68 ± 0.87	91.31 ± 1.52	**93.06 ± 2.12**
	P40Pro	90.87 ± 7.36	92.30 ± 4.04	90.43 ± 6.82	**92.61 ± 6.53**
	GoPro	92.95 ± 3.41	92.23 ± 4.91	**94.53 ± 2.96**	93.37 ± 4.29
#Set3	Xtion	86.72 ± 6.39	90.42 ± 4.59	90.83 ± 4.17	**92.25 ± 3.09**
	Iphone	91.33 ± 1.91	92.89 ± 2.44	87.63 ± 1.16	**93.68 ± 2.16**
	Phone	90.75 ± 6.87	92.05 ± 5.75	89.47 ± 6.54	**93.42 ± 6.11**
	GoPro	87.86 ± 3.77	89.44 ± 3.91	87.63 ± 4.71	**92.04 ± 4.19**
#Set4	Xtion	85.55 ± 6.89	88.83 ± 5.78	88.77 ± 4.50	**91.11 ± 5.56**
	Iphone	90.19 ± 1.79	88.87 ± 1.39	88.54 ± 3.27	**93.21 ± 3.55**
	P40Pro	82.37 ± 7.37	83.34 ± 8.05	81.81 ± 4.44	**87.01 ± 6.91**
	GoPro	90.03 ± 5.47	91.03 ± 4.46	91.20 ± 4.52	**93.75 ± 4.50**
#Set5	Xtion	87.66 ± 5.39	89.89 ± 5.45	89.69 ± 4.19	**92.47 ± 3.64**
	Iphone	92.86 ± 5.45	93.80 ± 1.39	88.49 ± 1.85	**94.78 ± 3.10**
	Phone	91.28 ± 6.81	**94.64 ± 3.79**	90.62 ± 6.25	93.98 ± 5.26
	GoPro	92.26 ± 3.90	91.03 ± 4.46	93.17 ± 3.55	**95.87 ± 2.99**
Mean		88.59 ± 4.52	90.08 ± 4.19	88.85 ± 4.12	**92.08 ± 4.24**

## Data Availability

Not applicable.

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
