# Peer review of "Cross-Domain Indoor Visual Place Recognition for Mobile Robot via Generalization Using Style Augmentation"

_sensors, 2023, doi:10.3390/s23136134_

Round 1
Reviewer 1 Report
This work proposes an algorithm for multi-domain visual recognition of a place indoors. It is based on a convolutional neural network, style randomization. A domain change is defined as a camera model change. A scene classification mechanism and improved performance of models based on synthetic and real data from various domains are suggested.
The paper's contribution to existing knowledge in this research field is well justified. The paper needs to contribute more; the following points can improve the manuscript.
1. It does not clearly show your innovations and contributions. Please highlight your innovations.
2. A comparative study can be added to the Relevant work section in table form to show the recent efforts.
3. More details about network architecture and the complexity of the model should be provided.
4. Discuss the performance evaluation metrics mathematically.
5. The proposed method should be compared with more recent techniques.
6. Tabular data should be presented with graphs.
7. Proofreading is recommended. For example, in the abstract, "It based on a convolutional neural network, style randomization." It should be "It is based on a convolutional neural network, style randomization.”, etc.
Proofreading is recommended. For example, in the abstract, "It based on a convolutional neural network, style randomization." It should be "It is based on a convolutional neural network, style randomization.”, etc.
Author Response
This work proposes an algorithm for multi-domain visual recognition of a place indoors. It is based on a convolutional neural network, style randomization. A domain change is defined as a camera model change. A scene classification mechanism and improved performance of models based on synthetic and real data from various domains are suggested.
The paper's contribution to existing knowledge in this research field is well justified. The paper needs to contribute more; the following points can improve the manuscript.
- It does not clearly show your innovations and contributions. Please highlight your innovations.
The innovation of the work and the contribution of the work are presented in the subsection ‘Objective of the work’ (line 90). The issues of the work and the lack in the current literature were listed and the contribution was determined on their basis.
- A comparative study can be added to the Relevant work section in table form to show the recent efforts.
The limitation in showing recent efforts is the specific scenario and assumptions adopted in the work. The focus was on convolutional networks relevant to the robot application and their operation in a complete multi-domain environment. The focus was on the operation of the method for unknown domains in the context of the camera model.
- More details about network architecture and the complexity of the model should be provided.
The work added a diagram for the structure used, "ResNet-50 architecture after transfer learning" (line 296) and a description for other convolutional neural networks. The Features Extraction chapter describes other tested structures.
- Discuss the performance evaluation metrics mathematically.
Information about performance evaluation metrics has been added in the Experimental Results chapter. Standard approaches in the literature are used and are linked to relevant literature references. “In the work, standard assessment indicators were used, the basic ones being Accuracy, Precision, Recall, and F1-score, which represents information about misclassified rooms from the sequence.” (line 343)
- The proposed method should be compared with more recent techniques.
Various methods available in the literature were analyzed, however, during the comparative tests, methods that can be used in a robot moving in a new environment were selected, taking into account such limitations as the algorithm response time, possibly low computational complexity, and a small amount of training data. References to more recent techniques along with their discussion have been added to the work in the "Relevant work" chapter.
- Tabular data should be presented with graphs.
Additional box plots have been added to the table, presenting the results in an additional way. They show the results of each method.
Figures:
- ‘Diagram of method classification results for a dataset sequence’;
- ‘Diagram of method classification results for the different domains sequences of the data sets’;
- ‘Diagram of sequence classification results using cross-domain data’.
- Proofreading is recommended. For example, in the abstract, "It based on a convolutional neural network, style randomization." It should be "It is based on a convolutional neural network, style randomization.”, etc.
Thank you for your comments. The text has been corrected.

Reviewer 2 Report
The abstract should include some quantitative results to show method efficacity regarding literature.
All contributions of this study should be listed in Section 1.
The authors must describe in detail all image adquisition conditions to build a realible dataset and how color constancy was taken into account, camera calibration, blurring, denoising methods, etc. Next, which kind of images were considered for the transfer learning approach?
Image deformation or perspective issues were considered to build synthetic images?
Describe more deep the transfer learning approach, issues and inconvenients in developping new samples.
The authors are encourging to use the created datasets on real robotic probleas as navigation and classification to verify method effectiveness.
This method could be extended to cope with 3D visual navigation on mobile robots?
Some bibliographical reference are outdated please verify the latest constribuitions on the field.
The manuscript requires deep proofreading.
Author Response
The abstract should include some quantitative results to show method efficacity regarding literature.
Added relevant information in the abstract: “The method of obtaining average results for the proposed method at the level of 92.08\%. The result obtained by another research team was corrected.”. (line 12)
All contributions of this study should be listed in Section 1.
A contribution has been introduced in the "Objective of the work" section. (line 90)
The authors must describe in detail all image adquisition conditions to build a realible dataset and how color constancy was taken into account, camera calibration, blurring, denoising methods, etc. Next, which kind of images were considered for the transfer learning approach?
The "Multi-Domain dataset" section describes the properties of the collected images that have not undergone additional modifications. This was due to the assumptions of the work and the lack of additional preparations before the method came into contact with the cameras. The difficulty of the method was increasing and a multi-dimensional field of action was necessary.
Fragment: “The data were not subjected to additional modifications in terms of contrast, color saturation, blurring, and noise reduction-using methods. This was due to the need to test the method on raw data….”. (line 228)
The subchapter "Style randomization" defines the type of set modification for synthetic augmentation not based on new sequences.
Fragment: “The proposed algorithm from one image was extended to four new images with separate random domains….” For each image, randomly generate data with a changed domain based on the mechanism style randomization. The extended data set went to the network transfer learning process. (line 265)
Image deformation or perspective issues were considered to build synthetic images?
The method in the work focusses on the main textures depicted in the image. The change of perspective was mainly based on the layers and operations carried out on the CNN structure. In the future, the work will cover a wider spectrum of modifications, also in perspective.
Describe more deep the transfer learning approach, issues and inconvenients in developping new samples.
Extended description: “The proposed algorithm from one image was extended to four new images with separate random domains.” and in the chapter "Transfer learning". (line 265)
The authors are encourging to use the created datasets on real robotic probleas as navigation and classification to verify method effectiveness.
Thank you for your attention. At work, we wanted to focus on presenting and testing deep networks in their approach to multi-domain images. We plan to use the developed method in practical applications in further work, e.g. rooms navigation. The practical application of the developed methods will be examined in subsequent works and will combine the assumptions and effects of the presented method.
This method could be extended to cope with 3D visual navigation on mobile robots?
The work currently includes CNN network research based on color image texture. In the future, there are plans to use 3D depth camera information in the context of navigation and multi-domain data. However, these are further works due to their complexity and time-consuming nature.
Some bibliographical reference are outdated please verify the latest constribuitions on the field.
Thank you for your attention. Newer works relevant to the topic of the article have been added [25,26, 27,28].

Reviewer 3 Report
The submitted manuscript claims to present an algorithm for multi-domain visual recognition of a place indoors based on a convolutional neural network and style randomization. However, no algorithm is presented, nor a logical function. The English used in the paper is of poor quality and requires extensive corrections. The numerous grammatical errors, sentence structure issues, and lack of coherence throughout the text make it challenging to understand the scientific content and evaluate the validity of your research.
I have several concerns regarding the novelty, clarity, and significance of the contribution. Additionally, the overall methodology and experimental setup require further elaboration and validation to establish the robustness and generalizability of the proposed approach.
Specifically, it should be noted that the description of the convolutional neural network and style randomization is lacking in technical detail, making it difficult to assess the novelty and effectiveness of the proposed method. Moreover, the dataset used for evaluation appears to have limitations, particularly in terms of the diversity and representativeness of the real-world scenarios.
Furthermore, an important drawback of the paper is the limited comparison with existing state-of-the-art methods in multi-domain visual recognition and the absence of a comprehensive analysis of the results obtained. Additional experiments and thorough statistical analysis are necessary to support the claims made in the paper.
The English used in the paper is of poor quality and requires extensive corrections. The numerous grammatical errors, sentence structure issues, and lack of coherence throughout the text make it challenging to understand the scientific content and evaluate the validity of your research.
Author Response
The submitted manuscript claims to present an algorithm for multi-domain visual recognition of a place indoors based on a convolutional neural network and style randomization. However, no algorithm is presented, nor a logical function. The English used in the paper is of poor quality and requires extensive corrections. The numerous grammatical errors, sentence structure issues, and lack of coherence throughout the text make it challenging to understand the scientific content and evaluate the validity of your research.
Additional descriptions and schemes of the "Algorithm diagram for cross-domain visual room recognition" (line 184) algorithm have been introduced. Additionally, the text was proofread.
I have several concerns regarding the novelty, clarity, and significance of the contribution. Additionally, the overall methodology and experimental setup require further elaboration and validation to establish the robustness and generalizability of the proposed approach.
The innovativeness of the work and the contribution of work are presented in the subsection “Objective of the work”. Added additional information regarding the used evaluation indicators in the "Experimental Results" section (line 4331). More statistical information has been introduced with the help of boxplots.
Figures:
- ‘Diagram of method classification results for a dataset sequence’;
- ‘Diagram of method classification results for the different domains sequences of the data sets’;
- ‘Diagram of sequence classification results using cross-domain data’.
Specifically, it should be noted that the description of the convolutional neural network and style randomization is lacking in technical detail, making it difficult to assess the novelty and effectiveness of the proposed method.
Additional descriptions for CNNs have been added in the paper, sub-chapter "Features extraction" (line 285). The work added a diagram for the structure used, "ResNet-50 architecture after transfer learning" (line 294). Additional information has been added to the Styles randomization chapter.
Moreover, the dataset used for evaluation appears to have limitations, particularly in terms of the diversity and representativeness of the real-world scenarios.
The limitations of other works are described in the chapter "Relevant work" (line 153) and elements of the set that introduce novelty are presented. The chapter details the collected images. In "Objective of the work" (line 90) the article's contribution is highlighted.
Furthermore, an important drawback of the paper is the limited comparison with existing state-of-the-art methods in multi-domain visual recognition and the absence of a comprehensive analysis of the results obtained. Additional experiments and thorough statistical analysis are necessary to support the claims made in the paper.
The paper presents a combination of methods with four methods known from the literature. The focus was on convolutional neural networks and their handling. The article also uses a comparison of the method with the results obtained by other research groups. The works were supplemented with graphs presenting statistical information about the obtained results, see boxplot:
Figures:
- ‘Diagram of method classification results for a dataset sequence’;
- ‘Diagram of method classification results for the different domains sequences of the data sets’;
- ‘Diagram of sequence classification results using cross-domain data’.
The English used in the paper is of poor quality and requires extensive corrections. The numerous grammatical errors, sentence structure issues, and lack of coherence throughout the text make it challenging to understand the scientific content and evaluate the validity of your research.
The text has been corrected.

Reviewer 4 Report
-more explanation on : “The dataset processing operation .. to these tasks was used” p6,line 199 would be helpful.
-more information is needed for the improvement of score to 98.74% p8, line 267
-for the proposed VPR algorithm for multi-domain and cross-domain data a pseudocode or a table with the steps would be useful.
-check the phrase for: “In the case ..images” p11,line:324
Author Response
-more explanation on : “The dataset processing operation .. to these tasks was used” p6,line 199 would be helpful.
Relevant information has been added in the sentence (line 284) "The data set processing operation, such as archiving, augmentation, CNN training, and SVM model, were not performed on the robot. It was too computationally expensive in conjunction with the work of the robot. An external computer adapted to these tasks was used. The images were processed, the data normalized, and the networks were trained on the computer...”
-more information is needed for the improvement of score to 98.74% p8, line 267
Added additional information on the basis of which it was possible to improve the result: “An improvement in the score to 98.74% indicated not having to use data from the deployment environment. The ResNet-50 model used was subjected to transfer learning based on images from the extended set proposed in the work. There was no connection between the rooms registered in the training and test sets.”. (line 378)
-for the proposed VPR algorithm for multi-domain and cross-domain data a pseudocode or a table with the steps would be useful.
It was proposed to present the algorithm in the form of a diagram Figure "Algorithm diagram for cross-domain visual room recognition". (line 184)
-check the phrase for: “In the case ..images” p11,line:324
Thank you for your attention. This and other bugs have been fixed.

Reviewer 5 Report
The paper considers the problem of visual place recognition using multiple robots as well as multiple domains.
Generally speaking, the presentation of this topic is at very superficial level with many open issues. Currently, the work corresponds to a technical report, only. There are a lot of numbers and pictures. However, there is no system model, no system architecture, no performance analysis, no comparison with existing solutions, limits of the proposed solution, etc.
In a journal publication, the reader should be able to redo the experiment with the ingredients outlined in the paper, which is currently not possible. Moreover, the reader should be able to benchmark any system with that in the presented paper, which is not possible either.
Finally, the novelty of the paper with respect to exisitng works is unclear. There have been numerous papers for room recognition using multiple robots.
Detailed comments:
1) : Which dataset exactly was used to simulate multiple robots in multiple domains. The authors point in line 143 to some reference [28] where the reader may pick one of many datasets.
2) what are the sensors, the robot(s) were equipped with?
3) What were the environmental conditions?
4) what was the festure space? and why it was chosen that way? what is the benefit of doing so with respect to previous works.
5) Many abbrevations have been used without explaining them. For example, line 114 states "Networks such as GoogLeNet
[21], ResNet-50 [22] or VGG-16 [23] ...." What kind of networks (wireless, neural?) be specific, explain the abbreviations!
only small errors that can be easily fixed.
Author Response
The paper considers the problem of visual place recognition using multiple robots as well as multiple domains.
Generally speaking, the presentation of this topic is at very superficial level with many open issues. Currently, the work corresponds to a technical report, only. There are a lot of numbers and pictures. However, there is no system model, no system architecture, no performance analysis, no comparison with existing solutions, limits of the proposed solution, etc.
The "Algorithm diagram for cross-domain visual room recognition" (line 184) diagram shows the approach used in the work. Performance analysis based on specific CNN network structures are presented in the Table "Time of the feature vector extraction of the deep networks used" (line 325). Shows the application times of the selected networks. In the context of the other elements of the system - image reading and writing, the SVM model did not analyze their performance because they had a lower priority. The paper lists the limitations of using the approach in relation to other approaches.
In a journal publication, the reader should be able to redo the experiment with the ingredients outlined in the paper, which is currently not possible. Moreover, the reader should be able to benchmark any system with that in the presented paper, which is not possible either.
Thank you for your attention, unfortunately, the work on making a significant part of the collection available is time-consuming. It will be made available shortly for the need to reproduce the results.
Finally, the novelty of the paper with respect to exisitng works is unclear. There have been numerous papers for room recognition using multiple robots.
The work introduces elements that distinguish it from the standard approach in the visual navigation of robots. A description of these issues can be found in the "Objective of the work" (line 90) subchapter and the current deficiencies in the analyzed literature have been identified.
Detailed comments:
1) : Which dataset exactly was used to simulate multiple robots in multiple domains. The authors point in line 143 to some reference [28] where the reader may pick one of many datasets.
The work focuses on the set in the problem of room categorization. Information in table no. 2 and 3 of the cited article. “The subsets focused on the issue of room categorization in sequences and various conditions, e.g. blur, deblur….” (line 189)
2) what are the sensors, the robot(s) were equipped with?
The text introduced the information "Information from the robot's sensors was limited to visual information, i.e., a color camera. The subsets added were based on different camera models.” Detailed technical data about the robot can be found in the cited paper. Classification results for the data set in [55] (line 390) - Nao robot.
3) What were the environmental conditions?
In the subchapter "Multi-Domain dataset" there is information about the environment conditions in which the dataset was collected. (line 189)
4) what was the festure space? and why it was chosen that way? what is the benefit of doing so with respect to previous works.
The features that were used came from the last layers of the deep networks used. Information about a specific layer can be found in the table "Properties of the feature vector extraction of the deep networks used" (line 308). The selection resulted from obtaining the best result for each of the structures. Result obtained on the basis of experience and preliminary work with networks.
5) Many abbrevations have been used without explaining them. For example, line 114 states "Networks such as GoogLeNet
Added additional information about deep networks used. The use of abbreviations in the text has been limited by making the content more precise.

Round 2
Reviewer 1 Report
The authors have addressed most of my concerns. The paper can be accepted.
Author Response
Thank you for your reviews and valuable comments.

Reviewer 2 Report
The author have correctly addressed all my concerns.
A last proofreading is required. The actual version of the manuscript is technically sound.
Author Response
The author have correctly addressed all my concerns.
A last proofreading is required. The actual version of the manuscript is technically sound.
Thank you for your reviews and valuable comments. We made an additional correction. In addition, in the previous round, proofreading was performed by a professional company.

Reviewer 3 Report
The authors have improved the paper structure and the quality of the presentation.
The authors should include details about the humanoid robot and the Xtion camera. The reference [41] does not include such information, however there is a link to the dataset. I assume that the robot was Nao, but which version?
Is the new dataset going to be released as open-access?
The English still needs to be checked, especially the newly added content. For example, the following phrases do not sound right:
“However, there was a vulnerability to many difficulties in such matters as the visual recognition of places.”
“”In the work, the variant in which the camera with and without the robot moved around selected rooms was interesting.”
“the authors do not take into account the situation in which the model is prepared from a completely different domain”
“the base was sequences from nine different locations for the classification problem recorded by a humanoid robot”
“In the experiments, the approach was not additionally parameterized and modifications”
"A learned model is used and trained on new smaller or modified data."
"The prepared content was implemented on the robot before it was able to work."
and so on.
Author Response
The authors have improved the paper structure and the quality of the presentation.
The authors should include details about the humanoid robot and the Xtion camera. The reference [41] does not include such information, however there is a link to the dataset. I assume that the robot was Nao, but which version?
"Nao robot V4 had an Intel Atom Z530 @ 1.6 GHz processor, 1 GB RAM, and 8 GB Flash memory. The Xtion PRO LIVE camera was placed on it to capture images of the environment. The camera worked in VGA standard, resolution 640×480, 30 fps."
Is the new dataset going to be released as open-access?
Yes, in the prepared work, the entire set used in the work will be expanded with new elements and will be available to the public.
The English still needs to be checked, especially the newly added content. For example, the following phrases do not sound right:
Thank you for your reviews and valuable comments.
“However, there was a vulnerability to many difficulties in such matters as the visual recognition of places.”
Corrected.
“In the work, the variant in which the camera with and without the robot moved around selected rooms was interesting.”
Corrected.
“the authors do not take into account the situation in which the model is prepared from a completely different domain”
Corrected.
“the base was sequences from nine different locations for the classification problem recorded by a humanoid robot”
Corrected.
“In the experiments, the approach was not additionally parameterized and modifications”
Corrected.
"A learned model is used and trained on new smaller or modified data."
Corrected.
"The prepared content was implemented on the robot before it was able to work."
and so on.
In the first round, the article was sent for professional proofreading. The text has been corrected.

Reviewer 5 Report
Quality of the manuscript has significantly improved.
minor style errors
Author Response
minor style errors
Thank you for your reviews and valuable comments. In the first round, the article was sent for professional proofreading. In the second round, minor stylistic errors were additionally corrected.
